Comparative genomic analysis of the IDD genes in five Rosaceae species and expression analysis in Chinese white pear (Pyrus bretschneideri)

Su Xueqiang 1
Meng Tiankai 2
Zhao Yu 1
Li Guohui 1
Cheng Xi 1
Abdullah Muhammad 1
Sun Xu 1
Cai Yongping 1 ypcaiah@163.com
Lin Yi 1 linyi992547404@163.com
1 School of Life Science, Anhui Agricultural University , Hefei , China
2 School of Life Sciences and Technology, TongJi University , Shanghai , China
McCormick Sheila
Electronic publication date: 2019 Mar 26
Publication date: 2019
Volume: 7
Electronic Location ID: e6628
Received 2018 Oct 4; Accepted 2019 Feb 15
Copyright: © 2019 Su et al.
Copyright year: 2019
Copyright holder: Su et al.
License: This is an open access article distributed under the terms of the Creative Commons Attribution License, which permits unrestricted use, distribution, reproduction and adaptation in any medium and for any purpose provided that it is properly attributed. For attribution, the original author(s), title, publication source (PeerJ) and either DOI or URL of the article must be cited.
License URL: https://creativecommons.org/licenses/by/4.0/

Keywords: INDETERMINATE DOMAIN (IDD) genes, Chinese white pear, Phylogenetic analysis, Microsynteny, Lignin synthesis, SCW formation

Funding: National Natural Science Foundation of China #31640068 Anhui Provincial Natural Science Foundation #1808085QC79 Graduate innovation fund of Anhui Agricultural University #2018yjs-41 This study was supported by the National Natural Science Foundation of China (Grant #31640068), the Anhui Provincial Natural Science Foundation (Grant #1808085QC79) and the Graduate innovation fund of Anhui Agricultural University (Grant #2018yjs-41). The funders had no role in study design, data collection and analysis, decision to publish, or preparation of the manuscript.

==============================
The INDETERMINATE DOMAIN (IDD) gene family encodes hybrid transcription factors with distinct zinc finger motifs and appears to be found in all higher plant genomes. IDD genes have been identified throughout the genomes of the model plants Arabidopsis thaliana and Oryza sativa, and the functions of many members of this gene family have been studied. However, few studies have investigated the IDD gene family in Rosaceae species (among these species, a genome-wide identification of the IDD gene family has only been completed in Malus domestica). This study focuses on a comparative genomic analysis of the IDD gene family in five Rosaceae species (Pyrus bretschneideri, Fragaria vesca, Prunus mume, Rubus occidentalis and Prunus avium). We identified a total of 68 IDD genes: 16 genes in Chinese white pear, 14 genes in F. vesca, 13 genes in Prunus mume, 14 genes in R. occidentalis and 11 genes in Prunus avium. The evolution of the IDD genes in these five Rosaceae species was revealed by constructing a phylogenetic tree, tracking gene duplication events, and performing a sliding window analysis and a conserved microsynteny analysis. The expression analysis of different organs showed that most of the pear IDD genes are found at a very high transcription level in fruits, flowers and buds. Based on our results with those obtained in previous research, we speculated that PbIDD2 and PbIDD8 might participate in flowering induction in pear. A temporal expression analysis showed that the expression patterns of PbIDD3 and PbIDD5 were completely opposite to the accumulation pattern of fruit lignin and the stone cell content. The results of the composite phylogenetic tree and expression pattern analysis indicated that PbIDD3 and PbIDD5 might be involved in the metabolism of lignin and secondary cell wall (SCW) formation. In summary, we provide basic information about the IDD genes in five Rosaceae species and thereby provide a theoretical basis for studying the function of these IDD genes.

Introduction

Zinc finger proteins are transcription factors with a finger-like domain that are widely distributed in animals, microorganisms, and the plant kingdom (Miller, Mclachlan & Klug, 1985). In fact, zinc finger proteins are found in all plant eukaryotic lineages, which indicates that these proteins might have been derived from a common eukaryotic ancestor. Zinc finger proteins have been identified and functionally analysed in several plant species, such as Jatropha curcas (Shi et al., 2018), Oryza sativa (Zhang et al., 2018), Musa acuminata (Chen et al., 2014). One group of this large family of proteins, the INDETERMINATE DOMAIN (IDD) proteins, has a highly conserved ID domain (Colasanti, Yuan & Sundaresan, 1998), which contains typical C2H2 and C2HC zinc finger motifs (Wu et al., 2008). C2H2 zinc finger transcription factors, which are one of the most thoroughly studied transcription factor families (Agarwal et al., 2007; Wei, Pan & Li, 2016), contain tandem repeat segments of approximately 30 amino acids, all of which have a highly conserved amino acid sequence: (F/Y)-XC-X2-5-C-X3-(F/Y)-X5-psi-X2-H-X3-5-H (wherein C and H represent cysteine and histidine, respectively, X represents any amino acid, and psi represents a hydrophobic residue) (Parraga et al., 1988). The structure obtained from this particular sequence can bind to Zn2+ and fold to form a structure with two β-sheets and an α-helix. Zn2+ mainly plays a role in the linking of individual amino acid chains and is also crucial to the function of zinc finger proteins (Islam, Hur & Wang, 2009; Tian et al., 2010).

The IDD protein family is a special type of C2H2 zinc finger subgroup. These proteins have a highly conserved ID domain with an N-terminal nuclear localization signal (NLS) and four distinct zinc finger motifs (ZF1, ZF2, ZF3 and ZF4) (Colasanti et al., 2006; Kozaki, Hake & Colasanti, 2004). Two canonical zinc fingers are found in the IDD genes, and the encoded proteins also contain two unusual CCHC fingers, one of which is also associated with the zinc finger domain in the Saccharomyces cerevisiae SW15 protein (Kozaki, Hake & Colasanti, 2004). Among the species for which a complete genome-wide identification of the IDD gene family has been performed, Malus domestica has a higher number of IDD genes than Arabidopsis thaliana, O. sativa and Zea mays (Colasanti et al., 2006; Fan et al., 2017). The functions of some IDD genes, particularly those from A. thaliana (IDD genes with similar functions have also been reported in O. sativa and Z. mays), have been identified. The function of the IDD genes involves seed germination (AtIDD1, 2, ZmIDD9, ZmIDDveg9) (Feurtado et al., 2011; Yi et al., 2015), leaf and flower development (AtIDD4, 14, 15, 16) (Cui et al., 2013), flowering regulation (ZmID1, AtIDD8, OsID1) (Colasanti, Yuan & Sundaresan, 1998; Seo et al., 2011; Park et al., 2008), starch metabolism (AtIDD5) (Ingkasuwan et al., 2012), the regulation of plant gravitropism (AtIDD14, 15, 16, LPA1, OsIDD16, 18), root development and nitrogen metabolism (AtIDD3, 8, 9, 10) (Jeong et al., 2015; Coelho et al., 2018).

Pear, which belongs to the Rosaceae family, is widely cultivated throughout the world and is popular due to its unique flavour. China has a long history of eating pears, and the pears cultivated in China are mainly Pyrus bretschneideri, Pyrus pyrifolia, Pyrus sinkiangensis and Pyrus ussuriensis (Vavilov, 1951; Xue et al., 2018). “Dangshan Su” pear (Pyrus bretschneideri cv. Dangshan Su) is a diploid pear variety that originated in Dangshan County, Anhui Province, China, and due to its good quality and high medicinal efficacy (Hamauzu et al., 2007; Huang et al., 2010; Zhang et al., 2017), the fruit has a very high market value. The stone cells in pear are closely related to the flesh texture, and the content of stone cells is a key factor in determining the quality of pear fruit (Cai et al., 2010; Yan et al., 2014). However, few studies have investigated the IDD genes during fruit development, and the only results for Musa acuminata have been reported (Chen et al., 2014). In Musa acuminata, the expression of MaIDD is closely related to fruit development, whereas, in Malus domestica, the IDD gene also appears to play a role in regulating flower induction. In addition, some zinc finger proteins, such as the CCCH-tandem zinc finger protein IIP4 (Zhang et al., 2018), A. thaliana C3H14, C3H15 (Chai et al., 2015) and OsIDD2 in O. sativa (Huang et al., 2018), are involved in plant secondary cell wall (SCW) formation. Although the function of several IDD genes has been studied and their potential functions can be explained by bioinformatics, it remains unknown whether IDD genes participate in the regulation of flower induction, fruit development, lignin biosynthesis and SCW formation in Pyrus bretschneideri.

Among Rosaceae species, genome-wide identification of IDD genes has only been performed in Malus domestica (Fan et al., 2017). Currently, there remain many questions about the IDD gene family in Chinese white pear. In this study, we identified the IDD genes in five Rosaceae plants and performed a comparative genomic analysis. Woodland strawberry (Fragaria vesca) is a perennial herb with edible fruits that grows well in most parts of the Northern Hemisphere, and its genome-wide sequencing was first reported in 2011 (Shulaev et al., 2011). Japanese apricot (Prunus mume) has great ornamental value and in some Asian countries, the fruits of this tree are used for cooking and flavouring. Sweet cherry (Prunus avium), which originated in Europe, Anatolia, Maghreb and western Asia, is an important economic tree species (Tavaud et al., 2004), and a new report on the genome sequence of this species was published in 2017 (Shirasawa et al., 2017). In addition, black raspberry (Rubus occidentalis), a specialty fruit crop that is grown in the Pacific Northwest of the United States, has a unique flavour and rich nutritional value, and the whole-genome sequencing of this species was published in 2017 (VanBuren et al., 2016). These four species and Chinese white pear belong to the Rosaceae family and might be connected through evolutionary relationships. We compared the IDD gene family of Chinese white pear with those of the four above-mentioned species to help us obtain a more comprehensive understanding of the IDD gene family and to further predict and analyse the functions of the IDD gene family members in Chinese white pear.

Materials and Methods

Identification of IDD genes in five rosaceae plants

In this study, the Chinese white pear genome database was obtained from the Pear Genome Project (http://gigadb.org/dataset/10008). The genome databases of three Rosaceae plants (F. vesca, R. occidentalis, Prunus avium) were downloaded from genome database for Rosaceae (GDR) (https://www.rosaceae.org/). The genome sequence data of Prunus mume were downloaded from the Prunus mume genome project (http://gigadb.org/dataset/10008). DNATOOLS software was used to create a local database containing the amino acid sequences of the IDD genes in the five Rosaceae plants (Cao et al., 2016). To identify the IDD genes of these five Rosaceae species, we used the following methods. First, the A. thaliana (16) and Malus domestica (20) IDD gene sequences were collected for use as query sequences in a TBlastN search with the default E-value (Table S1). We compared these sequences with the local database sequences of the above-mentioned five Rosaceae species. Using the SMART online software program, the IDD candidate gene sequences initially screened by BLAST were then tested to determine whether they contained a zinc finger domain (http://smart.embl-heidelberg.de/) (Letunic, Doerks & Bork, 2012). Each candidate sequence containing a zinc finger domain was subsequently manually checked for an IDD domain, and the protein sequences lacking a full IDD domain and redundant sequences were discarded. We then isolated the candidate IDD gene sequences and used the ExPASy online site to investigate the molecular weights of the IDD proteins (http://web.expasy.org/protparam/) (Artimo et al., 2012). The subcellular localization of all IDD proteins was predicted using the WoLF PSORT website (http://www.genscript.com/wolf-psort.html) (Horton et al., 2007). Phtre2 website (http://www.sbg.bio.ic.ac.uk/phyre2) for protein three-dimensional structures prediction (Kelley et al., 2015). Gene ontology (GO) annotations analysis we use BLAST2GO software (Conesa et al., 2005).

Phylogenetic analysis

Sequence alignment of all IDD proteins was done using the ClustalW tool in MEGA 7.0 software. The phylogenetic tree was constructed with MEGA 7.0 software using the maximum likelihood (ML) (bootstrap = 1,000) (Kumar, Stecher & Tamura, 2016). The Malus domestica IDD gene sequence used in phylogenetic tree was derived from the results of Fan et al. (2017). The sequences of A. thaliana, O. sativa and Z. mays were obtained from the article by Colasanti et al. (2006).

IDD gene structures and conserved motif prediction

INDETERMINATE DOMAIN gene structures were compared using Gene Structure Server (http://gsds.cbi.pku.edu.cn) (Guo et al., 2007). The motifs of the IDD genes in these five Rosaceae species were analyzed using MEME online analysis software (http://meme.sdsc.edu/meme4_3_0/intro.html) (Bailey et al., 2015). Parameters for the conserved motif prediction were motif width greater than 6 and less than 200. The number of identified motifs was 20.

Chromosomal location, gene duplication and Ka/Ks ratio analysis

Five Rosaceae species chromosome start positions and other relevant information about the IDD genes were obtained from the public genome database. The chromosomal physical locations of the IDD genes of all five Rosaceae species were mapped using MapInspect software (http://mapinspect.software.informer.com) (Hu & Liu, 2011; Lin et al., 2011; Zhu et al., 2015). To define gene duplication events, we mainly relied on the following principles: (1) The similarity of the two gene-coding sequences was more than 80%. (2) If the two genes were located on the same chromosome and separated by at least 200 kb, we considered these two genes tandem-duplicated genes. (3) If the two genes were located on different chromosomes, they were called segmentally duplicated genes (Long & Thornton, 2001). DnaSP v5.0 software was used to calculate non-synonymous (Ka) and synonymous substitution (Ks) values and perform sliding window analysis. Parameters for sliding window analysis were window size 150 bp and step size nine bp (Librado & Rozas, 2009).

Microsynteny analysis and Chinese white pear IDD gene promoter Cis-acting element analysis

We obtained the promoter sequence of each IDD gene from the Chinese white pear genome database, including the DNA sequence of the initiation codon (ATG) located 1,500 bp upstream of each IDD gene. The online software Plantcare was used to analyze the cis-acting elements of the promoter region (http://bioinformatics.psb.ugent.be/webtools/plantcarere/html/) (Rombauts et al., 1999). The microsynteny of six Rosaceae species (we added the IDD genes of Malus domestica) was identified using the Multiple Collinearity Scan toolkit (MCSscanX) (Abdullah et al., 2018).

RNA extraction and qRT-PCR

The plant material used in this experiment was the “Dangshan Su” pear, which grows in the Yuanyichang agricultural park (Dangshan County, Anhui Province, China). Samples of current-year flowers, flower buds, stems, mature leaves and fruits were obtained from 40-year-old Pyrus bretschneideri cv. Dangshan Su trees. The fruits were picked 15 days after pollination (DAP), 39 DAP, 47 DAP, 55 DAP, 63 DAP, 79 DAP and 145 DAP, and the fruits obtained 39 DAP were used for the expression analysis in different organs. “Dangshan Su” pear buds were treated with gibberellin (GA) and sucrose. Specifically, on a clear morning, terminal buds on current-year spurs were sprayed (<5 cm) with GA (700 mg·L−1) and sucrose (20,000 mg·L−1) using a low-pressure hand wand sprayer. Some plant materials were collected immediately after spraying with GA and sucrose and stored at −80 °C for use as controls. Materials collected 2 h (H), 4 H, 6 H, 8 H, 12 H and 24 H After GA and sucrose treatment were stored at −80 °C for gene expression analyses and samples collected 0 h post-treatment (HPT), 2 HPT, 4 HPT, 6 HPT, 8 HPT, 12 HPT and 24 HPT were also collected. RNA was extracted from the plant materials (including fruits and various organs) using a plant RNA extraction kit from Tiangen (Beijing, China). Reverse transcription was performed using a PrimeScript™ RT reagent kit with gDNA Eraser (Takara, Kusatsu, Japan), and each reaction consisted of one μg of RNA. The qRT-PCR primers for the pear IDD genes were designed using Beacon Designer 7 software (Table S8). Each 20-μL qRT-PCR system consisted of 10 μL of SYBR® Premix Ex Taq™ II (2×) (Takara, Kusatsu, Japan), two μL of cDNA, 6.4 μL of water and 0.8 μL of PbIDD-F and PbIDD-R. The pear Tubulin gene (No. AB239680.1) (Wu et al., 2012) was used as an internal reference. The reaction procedure was performed following the instruction manual, and three repetitions were run for each sample. The relative expression levels were calculated using the 2−ΔΔCt method (Livak & Schmittgen, 2001).

Results

Identification, characterization and phylogenetic analysis of IDD genes

A total of 68 IDD proteins were identified and used for further analysis (Table 1; Table S1), which included 16 IDD proteins (PbIDD1-PbIDD16) in Chinese white pear. We also identified a total of 52 IDD proteins in the other four species, including 14 in F. vesca (FvIDD1-FvIDD14), 13 in Prunus mume (PmIDD1-PmIDD13), 14 in R. occidentalis (RoIDD1-RoIDD14) and 11 in Prunus avium (PaIDD1-PaIDD11). Although only 11 IDD proteins were identified in Prunus avium, the numbers of IDD proteins in the other species were very similar but fewer than the number of IDD proteins in Malus domestica (20). We used ExPASy software to calculate the physicochemical parameters of the IDD genes (Table S2). We identified 68 IDD genes with isoelectric points (pIs) greater than 7. In these Rosaceae species, the lowest pI value was 7.76 (PaIDD8), whereas the highest pI value was 9.45 (FvIDD4). The molecular weights of these IDD genes were quite different, ranging from 42.0 kDa for PaIDD10 to 79.6 kDa for PmIDD8. The numbers of amino acids in the IDD proteins were also quite different. PbIDD14 contained the fewest (only 381) amino acids, whereas, PmIDD8 had the largest number (728). The grand average hydropathicity (GRAVY) of all 68 IDD genes was less than 1, and the smallest GRAVY value (−0.914) was obtained for PbIDD15.

Table 1 Basic Information of IDD Gene in Pyrus bretschneideri.

Gene name	Gene ID	AA	KD	pI	GRAVY	Preditced subcellular localization	
PbIDD1	Pbr029706.1	489	51.5	8.17	−0.460	nucl	
PbIDD2	Pbr006167.1	535	55.7	8.98	−0.427	nucl	
PbIDD3	Pbr032492.1	533	58.0	8.98	−0.659	nucl	
PbIDD4	Pbr021137.1	553	60.4	9.31	−0.634	nucl	
PbIDD5	Pbr028264.1	526	57.4	8.89	−0.693	nucl	
PbIDD6	Pbr019853.1	537	59.1	8.67	−0.807	nucl	
PbIDD7	Pbr012193.2	567	61.7	8.92	−0.772	nucl	
PbIDD8	Pbr020403.1	524	57.3	9.07	−0.740	nucl	
PbIDD9	Pbr009954.1	465	50.9	9.19	−0.664	nucl	
PbIDD10	Pbr012907.1	465	51.1	8.99	−0.629	nucl	
PbIDD11	Pbr008330.1	464	51.0	9.35	−0.727	nucl	
PbIDD12	Pbr012192.1	607	64.7	9.13	−0.723	nucl	
PbIDD13	Pbr025606.1	426	47.2	9.25	−0.646	nucl	
PbIDD14	Pbr029012.1	381	42.7	9.36	−0.655	nucl	
PbIDD15	Pbr038802.1	502	56.4	9.02	−0.914	nucl	
PbIDD16	Pbr012170.1	492	54.4	8.93	−0.805	nucl	
Note:

The IDD genes of Pyrus bretschneideri identified in this study are listed.

A phylogenetic tree of 98 IDD proteins was constructed using the ML method (Fig. 1). Based on the phylogenetic tree, we identified four phylogenetic groups (groups 1–4). Group 1 had the most members and contained all six Rosaceae species and 44 IDD genes: Pyrus bretschneideri (8), F. vesca (7), Prunus mume (6), R. occidentalis (7), Prunus avium (6), Malus domestica (10). Groups 3, 2 included 21 and 14 IDD genes, respectively. Group 4 had the fewest members: two IDD genes in Pyrus bretschneideri, two IDD genes in F. vesca, two IDD genes in Prunus mume, two IDD genes in R. occidentalis, one IDD gene found in Prunus avium and no IDD gene in Malus domestica. ZmID1 was grouped into a class by itself, and most PbIDD genes were more tightly grouped with MdIDDs. In addition, the IDD genes of some Rosaceae species were closely related to those of A. thaliana and O. sativa: PbIDD3, PbIDD5, FvIDD3, RoIDD4, PmIDD3 and PaIDD4 were included in group 1 and were closely related to OsIDD2; FvIDD12, RoIDD9, PmIDD11, PaIDD5 and AtIDD3 and 8 are closely related, and AtIDD1 was included in group 3 and clustered with seven Rosaceae IDD genes. The results of GO annotations (Table S9) showed that all 68 IDD genes had the function of nucleic acid binding (GO:0003676).

Figure 1 Phylogenetic relationships and subfamily designations in IDD proteins from Pyrus bretschneideri, Fragaria vesca, Prunus mume, Rubus occidentalis, Prunus avium, Malus domestica, Arabidopsis thaliana, Oryza sativa.

Phylogenetic relationships and subfamily designations in IDD proteins from Pyrus bretschneideri, Fragaria vesca, Prunus mume, Rubus occidentalis, Prunus avium, Malus domestica, Arabidopsis thaliana, Oryza sativa and Zea mays. Groups 1–4 are represented by shades of red, green, blue and cyan, respectively.

Structural and conserved motif analysis of IDD proteins

To more comprehensively analyse the structural diversity of the IDD genes in the five Rosaceae species, we created exon-intron pattern maps of all 68 IDD genes (Fig. 2). A total of 25 of the 34 IDD genes in group 1 had three exons and the other six genes had two exons. Additionally, FvIDD7 (4), RoIDD6 (5) and PmIDD8 (6) had a large number of exons. The analysis of group 2 revealed that FvIDD11 had two exons, PbIDD7 had four exons and all the other members had three exons. Group 3 comprised a subfamily with a complex gene structure: 11 genes had four exons and four genes had three exons. All members of group 4 had a gene structure comprising three exons. The lack of high similarity among the 68 IDD genes allowed a better understanding of the conserved motifs in these IDD genes. We used MEME software to identify 20 conserved motifs (Fig. 2; Table S3). Motifs 2, 3, 1 and 4 encoded the ID domain, whereas the remaining motifs did not have functional annotations. As shown in Fig. 3, motifs 2, 3, 1, 4 represented the zinc fingers ZF1, ZF2, ZF3 and ZF4, respectively. The four conserved motifs constituted a conserved ID domain, whereas these four conserved motifs represented two C2H2 and two C2HC structures (ZF1 and ZF2 belonged to C2H2, and ZF3 and ZF4 belonged to C2HC). Two C2H2 and two C2HC structures were identified through a sequence alignment of the conserved ID domains of five species and the results showed that the ID domain was highly conserved. The cysteine (C) and histidine (H) residues in each zinc finger domain were conserved in each species, consistent with the results of previous studies (Figs. S1–S5). All 68 IDD genes had motifs 1–4, which were the most conserved. In addition, some members of the same group and the more closely related members had highly similar motif compositions (e.g., FvIDD7 and RoIDD6 in group 1; PmIDD10 and PaIDD9 in group 2). These genes were closely related and had exactly the same motif compositions. We also identified certain group-specific motifs, such as motifs 12, 17, 20 in members of group 4 and motif 19 in some members of group 1.

Figure 2 Predicted Pyrus bretschneideri, Fragaria vesca, Prunus mume, Rubus occidentalis and Prunus avium IDD protein conserved motifs and exon-intron structures.

(A) Gene structures of the IDD genes. Black wedge indicates exon, black line indicates intron and red wedge indicates UTR. (B) Distribution of MEME motifs in IDD genes. (C) The color and corresponding number of each motif box.

Figure 3 Conserved ID domain composition.

All 68 IDD genes had a characteristic ID domain. The ID domain consists of four zinc fingers (Z1, Z2, Z3 and Z4). Alignment analysis of the 68 IDD gene sequences using the ClustalW tool in MEGA 7.0 software. These domain diagrams were plotted using the online WebLogo tool.

We also identified a NLS in the N-terminal border of these IDD genes (Fig. S6). According to previous studies, this NLS is usually KKKR or KRKR (Colasanti et al., 2006). Not all members in the five Rosaceae species, PbIDD13, 14, FvIDD10, 11, 12 and PmIDD11, have a NLS. In addition to the conserved IDD domain, we also found two C-terminal domains (MSATALLQKAA and TRDFLG), which are encoded by motifs 6, 7, respectively, in some members (Fig. 2; Figs. S1–S5). All nine members of group 4 lacked both motifs 6 and 7. Among the members of group 1, PaIDD10 did not contain motif 6, and PbIDD14 and PmIDD8 lacked motif 7. All members of groups 2, 3 contained these two motifs. The lack of these two conserved sequences at the C terminus of some IDD genes might explain why the members of group 4 were clustered into a single category and might also underlie the differences among the functions of these proteins (Fig. 1).

Chromosomal location and duplication events of IDD family genes in rosaceae

Based on the complete genome sequences of the five Rosaceae species, the exact chromosomal locations of all 68 IDD genes were identified (Fig. 4). In Pyrus bretschneideri, the 16 IDD genes were distributed on chromosomes 3, 4, 8, 9, 11, 12, 14, 15, 16, 17. In F. vesca, the IDD genes were distributed on chromosomes 1, 2, 3, 4, 5, 6. In R. occidentalis, the IDD genes were mainly distributed on chromosomes 2, 4 and 6. Several other IDD genes were distributed on chromosomes 1, 3 and 5. In Prunus mume, except PaIDD5, which could not be mapped to any chromosome, the IDD genes were distributed on chromosomes 1, 2, 4, 6, 7, 8. However, in Prunus avium, four IDD genes were distributed on chromosome 1 and the remaining 7 IDD genes were distributed on chromosomes 3, 5, 6, 7, 8. Only four pairs of duplicated genes were identified in pear (Fig. 4). All duplicated genes identified in Pyrus bretschneideri were segmental duplications.

Figure 4 Chromosomal locations of five Rosaceae species.

Chromosomal locations of IDD genes in Pyrus bretschneideri (A), Fragaria vesca (B), Rubus occidentalis (C), Prunus mume (D) and Prunus avium (E). Duplicated gene pairs are connected with colored lines.

To clarify the driving forces of gene duplication and explore the impact of these genes on evolutionary processes, we calculated the Ka, Ks and Ka/Ks ratio for the four duplicated gene pairs in Pyrus bretschneideri. Ka/Ks = 1 is the cut-off value that indicates neutral selection, Ka/Ks < 1 represents negative selection and Ka/Ks > 1 represents positive selection (Bitocchi et al., 2017). For all four duplicated gene pairs identified in Pyrus bretschneideri, the Ka/Ks values were less than 0.2726 (Table S4). In the evolutionary processes of genes, positive selection may be overshadowed by strong negative selection (Han et al., 2016). Therefore, to comprehensively explain the selection pressure of IDD genes, we performed a sliding window analysis of the four pairs of IDD paralogues in pears (Fig. S7). Most coding site Ka/Ks ratios were less than one, with exceptions for one or several distinct peaks (Ka/Ks > 1). The Ka/Ks ratios of the conserved IDD domains were less than 1.

Microsynteny and evolutionary analysis of the IDD gene family

For the identification of interspecific microsynteny, we visualized the location of homologous and orthologous genes. A total of 30 orthologous gene pairs were identified in five Rosaceae species (Fig. 5; Table S5) and these included one, eight, three, six and 12 collinear gene pairs between Pyrus bretschneideri and F. vesca, Prunus mume, R. occidentalis, Prunus avium, Malus domestica, respectively. Notably, PbIDD2 also showed a collinear relationship with the members of the MdIDD, PmIDD and PaIDD families. In contrast, two IDD genes from pear matched one pair of Prunus mume or R. occidentalis genes. For example, PbIDD9 and PbIDD10 were orthologous to PmIDD9, whereas PbIDD9 and PbIDD10 were orthologous to RoIDD7.

Figure 5 Microsynteny of regions among Pyrus bretschneideri, Fragaria vesca, Prunus mume, Rubus occidentalis, Prunus avium and Malus domestica.

The chromosome numbers are indicated by differently colored boxes and are labelled by Pb, Fv, Pm, Ro, Pa and Md. The differently colored boxes also represent the sequence lengths of chromosomes in megabases. The black line indicates the syntenic relationship among the IDD regions.

However, we did not identify any orthologous gene pairs between Pyrus bretschneideri and O. sativa, Z. mays and A. thaliana. We also compared the other four Rosaceae species with these three species, and the results did not reveal any orthologous gene pairs. To further study the evolution of IDD genes, we collected the IDD genes from four monocotyledons (M. acuminata, Sorghum bicolor, Z. mays and O. sativa) and two dicotyledons (Vitis vinifera and A. thaliana). A phylogenetic tree of the genomes of these 12 species was obtained to perform an evolutionary analysis of the IDD gene family (Fig. S8A). Several Rosaceae species exhibit a closer relationship and are more distant from monocots. This result is basically consistent with the results of the phylogenetic tree constructed from all IDD genes of 12 species (Fig. S8B). The results identified 12 classes (classes I–XII). The Pyrus bretschneideri and Malus domestica, Prunus avium and Prunus mume IDD genes often cluster into one class, and the IDD genes of monocotyledons are closely related (i.e., classes X and XI are only composed of IDD genes of monocotyledons). ZmID1, OsID1 and SbID1 are closely related, but we did not identify an orthologous gene for the ID1 gene in several other species. Class III contained all species except Malus domestica, and the IDD genes in this subfamily did not have complete MSATALLQKAA and TRDFLG domains.

Analysis of Cis-acting elements in the promoters of the IDD genes in Chinese white pear

Plant IDD genes may be involved in multiple biological processes, and hormones affect their expression. We predicted the cis-acting elements of 16 Chinese white pear IDD gene promoters (Fig. 6; Table S7). A total of 13 IDD genes (except PbIDD7, 9, 10) contained at least one G-Box, which is a light-responsive element, indicating that the expression of these genes may be regulated by light. Eight Pyrus bretschneideri IDD genes had HSE, and 11 members contained MBS components. TC-rich repeats, which are cis-acting elements involved in defence and stress responsiveness, were identified in PbIDD1, 6, 7, 9, 10, 12, 14 and 16. In addition, there were many cis-acting elements related to the responses to hormones, including responses to Abscisic acid response element (ABRE), methyl jasmonate (MeJA) (CGTCA-motif, TGACG-motif), GA (P-box, GARE-motif), auxin (TGA-element), and salicylic acid (TCA-element). The auxin-related TGA-element was identified in only five members (PbIDD1, 2, 4, 8, 13), which was the least number of elements components found. A total of 41 Cis-acting elements associated with MeJA were found in nine members, and these were the most abundant Cis-acting elements. Cis-acting elements responding to GA were the most widespread, and twelve of the Pyrus bretschneideri 16 IDD genes had this element. The related Cis-acting elements of abscisic acid and salicylic acid were identified in 10 and nine members, respectively. We found that PbIDD1 has Cis-acting elements related to the above five hormones, whereas PbIDD9 only has cis-acting elements that are responsive to MeJA. Of the 16 pear IDD genes, 12 contained 19 CGTCA-motifs and TCA-elements appeared in the promoter regions of 10 members, for a total of 22 genes. In addition, the CAT-box and the CCGTCC-box were found in 11 members (PbIDD1, 2, 3, 4, 5, 6, 7, 8, 9, 12, 14); these are Cis-acting elements related to meristem expression.

Figure 6 Promoter Cis-elements of the 16 PbIDDs.

Potential Cis-elements in the 5′ regulatory sequences of the 16 PbIDDs.

Expression characteristics of Chinese white pear IDD genes

To obtain a more in-depth understanding of the Chinese white pear IDD gene family, we investigated the expression patterns of PbIDDs in different organs (Fig. 7). No PbIDD14 expression was detected in any of the organs of Chinese white pear. Among the remaining 15 members, PbIDD4, 9, 11 were highly expressed in buds but showed extremely low expression in other organs. PbIDD1 and 10 were mainly expressed in flowers. PbIDD2, 6, 8 and 15 were mainly expressed in fruits, flowers and buds. Among these four genes, PbIDD2 and 8 showed the highest expression levels in buds, whereas PbIDD6 and 15 exhibited the highest expression levels in fruit. PbIDD5, 12, and 16 were detected in multiple organs, but the main difference in the expression of these genes was found in the leaves: (1) PbIDD12 was highly expressed in leaves; (2) although PbIDD5 was expressed in all five organs, its lowest expression was detected in the leaves; and (3) PbIDD16 showed almost no expression in leaves. PbIDD3, 7 and 13 were mainly expressed in the fruit of Chinese white pear, and PbIDD3 and 7 showed almost no expression in other organs.

Figure 7 Expression patterns of IDD genes of Chinese white pear in different organs and in fruit at different developmental stages.

Expression patterns of IDD genes in Chinese white pear in different organs (A–O). Expression patterns of IDD genes in Chinese white pear at different developmental stages (P–DD). *significant difference at P < 0.05, **significant difference at P < 0.01.

We investigated the expression patterns of 16 IDD genes in Chinese white pear fruit at seven developmental stages (Fig. 7). The expression level of PbIDD1 peaked at 39 DAP and was low during the other six fruit developmental stages. The expression peaks of PbIDD2, 7, 8, 11, 13, 16 also appeared at 39 DAP, but high expression of these genes was also detected at the early developmental stages (15 and 47 DAP). At the late developmental stages of fruit (79 and 145 DAP), these genes were expressed at a moderate level. PbIDD6, 15 exhibited a similar expression pattern as the above-mentioned genes, with the exception that PbIDD6, 15 were detected at the early developmental stages of fruit but were expressed at extremely low levels at the late developmental stages of fruit. The expression level of PbIDD4 was higher only at the early stage of fruit development (15–55 DAP) and at 79 DAP. PbIDD9 had a high transcription level at 63 and 145 DAP, and PbIDD12 was mainly expressed at 79 DAP. PbIDD10 had two expression peaks, one at 39 DAP and another at 55 DAP. PbIDD3 and 5 had the same expression pattern: these two genes were mainly expressed at the early stage of fruit development, and their expression level then gradually decreased.

Gibberellin and sugar response pattern analysis of PbIDDs

Chinese white pear PbIDD2, 5, 6, 8, 9, 12, 16 are mainly expressed in bud and flower. We treated the bud of Chinese white pear with exogenous GA and sucrose to investigate the expression pattern of PbIDD2, 5, 6, 8, 9, 12, 16 under GA and sucrose treatment (Fig. S9). Under exogenous GA treatment, PbIDD2, 5, 6, 8, 9, 12, 16 showed three different expression patterns (activation, inhibition and invariance). There was no significant change in the transcriptional level of PbIDD5, 9 under exogenous GA treatment. The expression levels of PbIDD2, 8 continued to decrease after exogenous GA treatment, indicating that the expression of PbIDD2, 8 was inhibited by GA. PbIDD6, 12 showed a significant increase in expression after GA treatment (reaching a peak at 3HPT); thereafter, the expression level gradually decreased to untreated levels. PbIDD16 reached the maximal expression levels at 2, 4HPT. Overall, the expression of PbIDD6, 12, 16 showed activation by GA. The transcription levels of PbIDD2, 8 increased gradually after sucrose treatment in Chinese white pear bud. The expressions of PbIDD5, 12, 16 were not always induced by sucrose. The lowest expression levels of PbIDD5, 12 were found at 4HTP, while the lowest expression level of PbIDD16 was observed at 1HTP. The expressions of PbIDD6, 9 were inhibited by sucrose.

Discussion

IDD gene family encode hybrid transcription factors that have four zinc finger structures and one nuclear positioning signal. IDD proteins play important roles in regulating plant flowering, development, stress resistance, secondary metabolism and other processes (Wong & Colasanti, 2007; Matsubara et al., 2008). The plant IDD gene can also regulate the expression of the key enzyme genes of lignin synthesis, which affects the biosynthesis of lignin and regulates the biosynthesis of SCW formation (Huang et al., 2018).

In this study, we identified 68 IDD genes in five Rosaceae species (Table 1; Table S2). The pI values of all IDD protein were greater than 7, indicating that all these proteins are alkaline proteins. The proteins were all hydrophilic, as demonstrated by the finding that their GRAVY values were less than 1. Among the Rosaceae species, Pyrus bretschneideri had the second greatest number of IDD gene family members after Malus domestica (Fan et al., 2017), and this difference might be traced back to the origin of the IDD gene family, during which time the numbers of IDD genes in Pyrus bretschneideri and Malus domestica were different. Previous studies have shown that plant whole-genome duplication (WGD) and chromosome rearrangement alter the sequences of genes, and this alteration is accompanied by widespread gene loss (Adams & Wendel, 2005; Tang et al., 2008). The common ancestor of Rosaceae species has nine chromosomes (Shulaev et al., 2011; Wu et al., 2013). Pyrus bretschneideri and Malus domestica both experienced two WGDs 130 million years ago (Mya) and 30–45 Mya, but the number of chromosomes was only 17 (Jaillon et al., 2007; Guo et al., 2013; Wu et al., 2013). This finding indicates that the nine chromosomes of Rosaceae species ancestors experienced doubling, breaking and fusion during the long process and ultimately formed the 17 chromosomes of Pyrus bretschneideri and Malus domestica. During this process, the genome of a species might become unstable and prone to gene replacement, leading to chromosome rearrangement and gene loss (Paterson, Bowers & Chapman, 2004). It is possible that the IDD genes in Pyrus bretschneideri were lost during this process, which explains why the number of IDD genes in Pyrus bretschneideri was less than that in Malus domestica. The phylogenetic tree divided 68 IDD genes into four groups (Fig. 1). ZmID1 comprised a separate class in the phylogenetic tree, and we did not identify a closely related gene (Fig. 1). At present, ZmID1 homologous genes have been identified only in O. sativa and S. bicolor, which belong to the same family of Gramineae (Colasanti et al., 2006). This finding might indicate that the ID1 gene is unique to Gramineae plants. Interestingly, five Rosaceae species had at least one IDD gene present in each clade, and this finding had strong bootstrap support. These results showed that the rapid duplication of IDD genes occurred before the divergence of these Rosaceae species.

The gene structure and conserved motif composition might be closely related to the diversity of gene functions (Swarbreck et al., 2008). Genes in the same subfamily tend to have very similar gene structures, which indicates that these genes might have similar functions (Fig. 2). For example, PbIDD12 and PmIDD10 in group 2 had the same gene structure (three exons and two introns), and their exon lengths were almost the same. Furthermore, based on the MEME analysis (Fig. 2), we found that members of the same subfamily had roughly the same conserved protein motifs, but some differences in motif composition were found between different subfamilies. An N-terminal NLS is absent in many Rosaceae IDD genes (Fig. 6). All IDD genes in A. thaliana have a NLS. OsIDD10, which belongs to the O. sativa IDD gene family, also lacks a NLS (Colasanti et al., 2006). The results indicate that the lack of NLS sequence in some members of the IDD gene family might be due not to the mutation or deletion of this entire characteristic sequence but rather to an extensive deletion of the N-terminal sequence. The ancestors of the special IDD genes in these Rosaceae species might have appeared after the differentiation of these Rosaceae species from the common ancestors of A. thaliana, and some of the N-terminal sequences, which includes the NLS, were lost after the evolutionary process. In addition to the highly conserved ID domain, we found two small domains (Figs. S1–S5) in the C-terminal region (except in members of group 4). The MSATALLQKAA and TRDFLG motifs are conserved regions outside the ID domain but are not characteristically conserved domains of IDD genes. Obviously, the members of group 4 lost these two small domains during evolution, which resulted in the generation of functional differences. We found that group 4 contained genes in A. thaliana (AtIDD15) and O. sativa (OsIDD14). AtIDD14, 15, 16 synergistically regulate spatial auxin biosynthesis by direct targeting YUCCA5 (YUC5), TRYPTOPHAN AMINOTRANSFERASE of ARABIDOPSIS1 (TAA1) and PINFORMED1 (PIN1) and thereby controlling aerial organ morphogenesis and gravitropic responses in plants (Cui et al., 2013). OsIDD14 is the functional orthologue gene of AtIDD15 and plays a role in the regulation of shoot gravitropism in O. sativa. Members of this group might have evolved to have functions in regulating the geotropism of plant organs. However, this group does not contain Malus domestica IDD genes, which indicates that MSATALLQKAA and TRDFLG domains might have been preserved during the evolutionary process or that IDD genes do not play a role in the regulation of organ geotropism in Malus domestica.

The functional divergence of genes often occurs after gene duplication and is accompanied by the introduction of new functions or the loss of original functions (Chao et al., 2017). In addition, gene duplication events are the main driving force for gene family expansion and thus constitute an important method used by plants to adapt to changing climates and environments (Tang et al., 2016). Gene duplication events have been identified in multiple gene families, such as the PKS gene family in Gossypium hirsutum (Su et al., 2017), the HSF gene family in Sesamum indicum (Dossa, Diouf & Cisse, 2016), and the WRKY gene family in Musa acuminate (Goel et al., 2016). A total of four PbIDD gene pairs were identified as duplications in this study, and these duplicated gene pairs were all created by segmental duplication (Table S4). Four pairs of segmentally duplicated genes were also identified in Malus domestica, verifying the reliability of our results. However, we did not identify gene duplication events in F. vesca, Prunus mume, R. occidentalis or Prunus avium. This finding might be due to the fact that F. vesca, Prunus mume, R. occidentalis and Prunus avium underwent only one WGD, but Pyrus bretschneideri had a recent WGD in addition to an old WGD (Zhang et al., 2012; Wu et al., 2013). The four pairs of duplicated genes that appeared in pears exerted a very positive effect on the expansion of the IDD gene family and contributed to the functional diversification of IDD genes. We then performed a sliding window analysis of the four pairs of segmentally duplicated genes in Chinese white pear, and the results showed that these genes experienced intense purifying selection to maintain their functional stability.

We did not identify the orthologous gene pair of IDD genes between several dicotyledons and monocotyledons. Therefore, we preliminarily hypothesized that IDD genes might have appeared after the differentiation of the common ancestors of these species differentiated. After the differentiation of monocotyledons and dicotyledons, two canonical zinc fingers and two unusual CCHC fingers formed a special zinc finger domain (ID domain), which defines IDD genes in all plants. Based on the results, we drew a hypothetical IDD genes evolutionary model map of Rosaceae species (Fig. S10). The ancestors of dicotyledons evolved through complex evolutionary processes to produce Rosaceae species. After an old WGDs, F. vesca, R. occidentalis, Prunus avium and Prunus mume formed the current IDD gene family. Malus domestica and Pyrus bretschneideri formed the IDD gene family after two WGDs. Subsequently, Pyrus bretschneideri and another Rosaceae species differentiated into two IDD gene types (with or without the MSATALLQKAA and TRDFLG domains). The IDD gene lacking these two domains may have derived the function of regulation of plant organ geotropism. Malus domestica did not detect such genes, perhaps because Malus domestica does not rely on IDD genes to participate in the regulation of organ geotropism.

Previous research has shown that IDD genes are most highly expressed in leaves and buds (Fan et al., 2017), but the highest pear IDD gene expression was detected in fruit, flower and bud organs. PbIDD7 and MdIDD19 represent a pair of homologous genes that are closely related. MdIDD19 was mainly expressed in the leaf, fruit and bud, whereas PbIDD7 was mainly expressed in fruit and bud organs and showed almost no expression in leaves (Fig. 7). The difference between the two homologous gene expression patterns was mainly concentrated in the leaf expression pattern. Many IDD genes are associated with leaf development and photosynthesis. However, we did not identify cis-acting elements related to the light response in the promoter region of PbIDD7 (Fig. 6), whereas abundant light-responding elements have been found in the promoter region of MdIDD19 (Fan et al., 2017). This finding might be the main reason for the large difference in the expression patterns between PbIDD7 and MdIDD19 in leaves. We found that PbIDD4 showed a high transcription level only in buds but exhibited a very low expression level in fruits and other organs (Fig. 7). This gene might not be the main gene involved in various physiological metabolism in fruit but plays an important role in the development of buds. Interestingly, we found that PbIDD2, 5, 6, 8, 9, 12, 16 showed higher transcription levels in flowers and buds, although the expression in buds was higher than that in flowers. In A. thaliana, O. sativa, Z. mays, IDD genes, such as AtIDD8, ZmID1, and OsID1, are involved in plant flower induction and are regulated by hormones and sugars (Matsubara et al., 2008; Wu et al., 2008; Jeong et al., 2015). GA can regulate plant flowering by inhibiting flowering genes (Zhang et al., 2016; Galvao et al., 2012). The promoter regions of PbIDD2, 6, 8, 12, 16 contain the GA-response cis-acting element. Chinese white pear buds were treated with exogenous GA, and the resulting expression patterns of PbIDD2, 5, 6, 8, 9, 12, 16 were detected, which revealed that these genes were mainly expressed in the buds and flowers of Chinese white pears (Fig. S9). PbIDD5, 9 were not induced by GA, probably because these genes do not contain GA-related cis-acting elements. After exogenous GA treatment, the expression level of PbIDD2, 8 decreased continuously, indicating that these genes were inhibited by GA. However, the expression levels of these two genes increased gradually after sucrose treatment, indicating that PbIDD2, 8 were induced by sucrose. We also identified a large number of light-responsive cis-acting elements in the promoter regions of PbIDD2, 8, which suggested that these two genes might respond to photoperiodic signals. Similar results have been found in Malus domestica: MdIDD7 is inhibited by GA in response to sucrose (Fan et al., 2017). In summary, the results suggested that PbIDD2 and 8 might participate in the GA flowering pathway during physiological flower bud differentiation and regulate the plant flowering process in response to sugar and photoperiod signals. PbIDD1 and 10 were mainly expressed in the flowers of Chinese white pear, probably because these two genes are involved in the synthesis of bioactive substances in flowers.

The stone cells and lignin in “Dangshan Su” pear fruit are mainly formed during the period from 15 to 63 DAP and reach their peak levels at 55 DAP (Zhang et al., 2017). The expression analysis of these 16 PbIDDs at seven developmental stages of fruit showed that the expression pattern of any one gene was not consistent with the trends obtained for the “Dangshan Su” pear fruit stone cell and lignin contents (Fig. 7). However, there were two additional specific genes (PbIDD3 and PbIDD5) and the expression patterns of these two genes showed completely opposite trends compared with the accumulation of “Dangshan Su” pear stone cells and lignin. PbIDD3 and 5 showed extremely high expression levels at 15 DAP. From 15 to 47 DAP, which is a stage characterized by the massive accumulation of lignin and stone cells, the expression of these two genes gradually decreased. At the peak of lignin accumulation (47, 55 and 63 DAP), PbIDD3 and 5 showed very low transcription levels, whereas at the late stage of fruit development (79 and 145 DAP), the expression levels of these genes increased slightly. According to the phylogenetic tree (Fig. 1), PbIDD3, PbIDD5 and OsIDD2 were the most closely related genes. In O. sativa, OsIDD2 regulates SCW formation while negatively regulating the expression of genes encoding key lignin synthesis enzymes, such as cinnamyl alcohol dehydrogenase (CAD), to inhibit lignin biosynthesis (Huang et al., 2018). In Chinese white pear, CAD is predicted to regulate the biosynthesis of fruit lignin. The expression pattern of PbCAD is exactly the same as the accumulation patterns of stone cells and lignin (Cheng et al., 2017). The expression patterns of PbIDD3 and PbIDD5 were opposite to those found for PbCAD. Moreover, PbIDD3 and PbIDD5 expression was low during the high expression period of PbCAD and was higher during the low expression period of PbCAD. In addition, PbIDD3 and PbIDD5 exhibited extremely high transcription levels in fruit. Subsequently, we predicted the three-dimensional structures of PbIDD3, 5 and OsIDD2 (Fig. S11). As shown in Fig. S11, the three-dimensional structures of PbIDD3, 5 exhibit high similarity with that of OsIDD2, indicating that these proteins might have similar functions and molecular mechanisms in regulating downstream gene expression. We speculate that PbIDD3, 5 are mainly expressed in fruit and have similar functions to OsIDD2 in regulating SCW formation in pear fruit cells and inhibiting lignin biosynthesis by inhibiting the expression of the genes encoding key lignin synthesis enzymes. In future studies, the overexpression and gene silencing of PbIDD3, 5 in “Dangshan Su” pear will be performed to determine their functions, and the interactions between PbIDD3, 5 and PbCAD will be explored through a yeast one-hybrid assay.

Conclusion

We identified 68 IDD genes in five Rosaceae species (Pyrus bretschneideri, F. vesca, Prunus mume, R. occidentalis and Prunus avium), which we systematically assessed by bioinformatic analysis. According to the phylogenetic tree, the 68 IDD genes were divided into four groups. In each class, we found that the structures of the genes and the compositions of the conserved motifs were very similar. Through a series of bioinformatics analyses, we explained the possible evolutionary patterns of IDD genes in these Rosaceae species. According to qRT-PCR, Chinese white pear IDD genes have high expression levels in fruit, flower and bud. Further, PbIDD2, 8 are mainly expressed in the buds of Chinese white pear and are responsive to the induction of GA and sucrose. That PbIDD3, 5 are may be involved in the regulation of SCW formation and lignin biosynthesis in Chinese white pear fruit. All in all, this study reveal the basic information of the five Rosaceae species’ IDD genes and predicts the potential functions of some pear IDD proteins. These results will provide an important theoretical basis for improving the quality of “Dangshan Su” pear.

Supplemental Information

Supplemental Information 1 Pyrus bretschneideri IDD protein sequence alignment.

Black underline indicates zinc finger domain (Z1, Z2, Z3 and Z4). Red triangle indicates a conserved C residue, and blue triangle indicates a conserved H residue. The yellow underline indicates the NLS sequence in the N-terminal region of the IDD gene. Green box means the MSATALLQKAA domain, and purple box indicates the TRDFLG domain.

Click here for additional data file.

Supplemental Information 2 Fragaria vesca IDD protein sequence alignment.

Black underline indicates zinc finger domain (Z1, Z2, Z3 and Z4). Red triangle indicates a conserved C residue, and blue triangle indicates a conserved H residue. The yellow underline indicates the NLS sequence in the N-terminal region of the IDD gene. Green box means the MSATALLQKAA domain, and purple box indicates the TRDFLG domain.

Click here for additional data file.

Supplemental Information 3 Prunus mume IDD protein sequence alignment.

Black underline indicates zinc finger domain (Z1, Z2, Z3 and Z4). Red triangle indicates a conserved C residue, and blue triangle indicates a conserved H residue. The yellow underline indicates the NLS sequence in the N-terminal region of the IDD gene. Green box means the MSATALLQKAA domain, and purple box indicates the TRDFLG domain.

Click here for additional data file.

Supplemental Information 4 Rubus occidentalis IDD protein sequence alignment.

Black underline indicates zinc finger domain (Z1, Z2, Z3 and Z4). Red triangle indicates a conserved C residue, and blue triangle indicates a conserved H residue. The yellow underline indicates the NLS sequence in the N-terminal region of the IDD gene. Green box means the MSATALLQKAA domain, and purple box indicates the TRDFLG domain.

Click here for additional data file.

Supplemental Information 5 Prunus avium IDD protein sequence alignment.

Black underline indicates zinc finger domain (Z1, Z2, Z3 and Z4). Red triangle indicates a conserved C residue, and blue triangle indicates a conserved H residue. The yellow underline indicates the NLS sequence in the N-terminal region of the IDD gene. Green box means the MSATALLQKAA domain, and purple box indicates the TRDFLG domain.

Click here for additional data file.

Supplemental Information 6 N-terminal region of the ID-domain shows the putative NLS sequence.

The yellow underline indicates the NLS sequence in the N-terminal region of the IDD gene.

Click here for additional data file.

Supplemental Information 7 Sliding window plots of duplicated IDD genes in Chinese white pear.

The grey shaded portion indicates conserved ID domain. The X-axis indicates the synonymous distance within each gene.

Click here for additional data file.

Supplemental Information 8 Two Phylogenetic tree of the 12 species genomes and IDD proteins from 12 species.

A phylogenetic tree of the 12 species genomes (A). Phylogenetic relationships and subfamily designations in IDD proteins from 12 species (B).

Click here for additional data file.

Supplemental Information 9 Expression modes of candidate PbIDD2, 5, 6, 8, 9, 12, 16 in Chinese white Pear buds treated with gibberellin (A-G) and sucrose (H-N).

*significant difference at P < 0.05, **significant difference at P < 0.01.

Click here for additional data file.

Supplemental Information 10 A hypothetical evolutionary model map of IDD genes.

Click here for additional data file.

Supplemental Information 11 Predicted three-dimensional structures of PbIDD3, 5 and OsIDD2.

OsIDD2 have been proven to be responsible for SCW formation and lignin biosynthesis.

Click here for additional data file.

Supplemental Information 12 Gene sequence list.

Click here for additional data file.

Supplemental Information 13 Basic information of IDD genes in four Rosaceae species.

The IDD genes of Fragaria vesca, Prunus mume, Rubus occidentalis and Prunus avium identified in this study are listed.

Click here for additional data file.

Supplemental Information 14 Detailed information of the 20 motifs in the 68 IDD proteins.

Click here for additional data file.

Supplemental Information 15 Ka/Ks analysis of the duplicated IDD paralogues from Chinese white pear.

Click here for additional data file.

Supplemental Information 16 Synteny data in five Rosaceae species.

Synteny data in Pyrus bretschneideri, Fragaria vesca, Prunus mume, Rubus occidentalis, Prunus avium, Malus domestica.

Click here for additional data file.

Supplemental Information 17 Promoter sequence of 16 IDD genes in Chinese white pear.

Click here for additional data file.

Supplemental Information 18 Complete information on Cis-acting elements of the 16 PbIDD genes.

Click here for additional data file.

Supplemental Information 19 Primer sequences used in qRT-PCR.

Click here for additional data file.

Supplemental Information 20 GO annotations analysis of all 68 IDD genes.

Click here for additional data file.

Supplemental Information 21 Raw data.

Click here for additional data file.

The authors are deeply grateful to Prof. Yongping Cai and Prof. Yi Lin, who provided the sample used in the study and very effective direction. The authors also thank Dr. Xu Sun, Dr. Muhammad Abdullah, Dr. Xi Cheng, Dr. Guohui Li, Dr. Yu Zhao for providing valuable suggestions and comments. Finally, author would like to thank Dr. Meng for his help in revising the paper.

Additional Information and Declarations

Competing Interests

Author Contributions

Data Availability

The authors declare that they have no competing interests.

Xueqiang Su conceived and designed the experiments, performed the experiments, contributed reagents/materials/analysis tools, prepared figures and/or tables, authored or reviewed drafts of the paper, approved the final draft.

Tiankai Meng analyzed the data, authored or reviewed drafts of the paper, modify our language problem.

Yu Zhao conceived and designed the experiments, performed the experiments, contributed reagents/materials/analysis tools, prepared figures and/or tables, authored or reviewed drafts of the paper.

Guohui Li analyzed the data, prepared figures and/or tables.

Xi Cheng performed the experiments, analyzed the data, contributed reagents/materials/analysis tools, prepared figures and/or tables.

Muhammad Abdullah performed the experiments, analyzed the data, contributed reagents/materials/analysis tools, prepared figures and/or tables.

Xu Sun analyzed the data, prepared figures and/or tables.

Yongping Cai conceived and designed the experiments, authored or reviewed drafts of the paper, approved the final draft.

Yi Lin conceived and designed the experiments, authored or reviewed drafts of the paper, approved the final draft.

The following information was supplied regarding data availability:

The raw measurements are available in the Supplemental Files.

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
