# Peer review of "Comparative genomic analysis of the IDD genes in five Rosaceae species and expression analysis in Chinese white pear (Pyrus bretschneideri)"

_PeerJ, doi:10.7717/peerj.6628_

## Round 0.1 · original submission · Major Revisions

Please address all the reviewer comments and have your manuscript edited for language.

Reviewer 1 ·

Basic reporting

Qverall, the results are informative. Because there are many grammatical errors in English writing, it is strongly recommended that manuscript needs to be polished by an English native speaker.

The logic was weaker in INTRODUCTION. One of the important content of this article is the identification of IDD genes from five Rosaceae species. However, only pear is introduced and no any introduction of other four Rosaceae species. You'd better briefly introduce these species and their genomes to give readers an idea of your research goals. Why did you do this? Functional comparison? Evolutionary comparison?

Experimental design

no comment

Validity of the findings

The functions of the IDD genes in pear are all speculated partially based on the expression levels. There is limited evidence showing that PbIDDs 3, 5 play a role in SCW formation and lignin biosynthesis in pear fruit and no evidence showing that PbIDDs 2, 8 are involved in the regulation of plant flowering induction in pear. Therefore, you should revise your conclusion.

Additional comments

The manuscript deals with the comparative genomic analysis of the IDD genes in five Rosaceae species and expression analysis in Chinese White Pear (Pyrus bretschneideri). In general, the results are informative. Although there are many grammatical errors in English writing, overall, the English of the manuscript is readable. However, it is strongly recommended that manuscript needs to be polished by an English native speaker. In addition, the functions of the IDD genes in pear are all speculated partially based on the expression levels. There is limited evidence showing that PbIDDs 3, 5 play a role in SCW formation and lignin biosynthesis in pear fruit and no evidence showing that PbIDDs 2, 8 are involved in the regulation of plant flowering induction in pear. Therefore, you should revise your conclusion.
There are other issues that need to be revised:
1) The authors identified IDD genes from 5 Rosaceae species belonging to different genera. The author used both common names and scientific names. However, common names of species are not consistent with scientific names which create big confusion. A common name fruit trees may include different species, like pear (Pyrus communis, Pyrus pyrifolia…). Fragaria vesca refers to woodland strawberry, not strawberry which is assigned to Fragaria × ananassa. Prunus mume is not a plum, but scientific name of Japanese apricot. Although Prunus mume is sometimes called Chinese plum, which is not popular. The real Chinese plum is usually assigned to Prunus salicina. So, please confirm whether you identified IDD genes from plum or from Japanese apricot. Rubus occidentalis refers to common name black raspberry. Therefore, please revise the common names of fruit trees used in your study. In most cases, it is recommended that only using scientific names to replace the common names.
2) Please use cultivar to replace variety. Variety is often used to refer to infraspecific taxon. While cultivar refers to the most basic classification category of cultivated plants.
3) The logic was weaker in INTRODUCTION. One of the important content of this article is the identification of IDD genes from five Rosaceae species. However, only pear is introduced and no any introduction of other four Rosaceae species. You'd better briefly introduce these species and their genomes to give readers an idea of your research goals. Why did you do this? Functional comparison? Evolutionary comparison?
4) –Line 68-69: you mentioned that ‘Dangshan Su’ had a high medicinal efficacy. What kinds of medicinal efficacy? Please cite references.
5) –Line 69: please use pear flesh to replace pear pulp.
6) –line 70: “they determine the quality of the pear fruit”? please specify the meaning of fruit quality. Stone cells may determine the fruit texture. However, fruit quality includes many aspects.
7) –Line 89-90: please reword “Among the Rosaceae species, only apples have had their IDD genes identified genome-wide”.
8) --Line 98, what does DAF mean? You should use the full name before the first abbreviation.
9) --Line 104 -105: “The genome databases of the other four Rosaceae plants (Strawberry, plum, Raspberry and Cherry) were downloaded from GDR”. However, no genome sequence data of Prunus mume can be found in GDR website. You should check carefully.
10) –Line 121: what is the meaning of “can from the study of Fan”?
11) --Line 122: Arabidopsis thaliana should be italic. In addition, all the gene name should be italic. Please check the text and revise.
12) –Line 134: ” (http://mapinspect.software.informer.com) (Niu et al., 2016)” Please cite original reference to replace Niu et al., 2016.
13) –Line 151: Suzhou County? Anhui Province?
14) --Line 191: please reword the sentence: “2 pear genes, 2 strawberry, 2 plum, 2 raspberry, 1 cherry, and no apple”.
15) In the section RESULTS, please avoid presenting any discussions and only describe your experimental data using past tense. Please remove sentence of line 183 to line 190, line 240 to line 242, line 245-246 line 259 to 2263, and so on.
16) –Line 271:Chinese pear should be replaced by Chinese white pear.
17) Please perform the significance analysis in Fig.7
18) In the section DISCUSSION, when you cite your experiment results, you should indicate which figure or table you refer to.
19) Please remove the repeated results in the discussion. For instance: Line 335-338.
20) --Line 367: ‘group4’ should be ‘group 4’. Please also check other parts of the text.
21) In section CONCLUSION, please revise “we believe that PbIDD2, 8 are involved in the regulation of plant flowering induction in pear and that PbIDD3, 5 are involved in the regulation of SCW formation and lignin biosynthesis in pear fruit”. In this study you don’t have solid evidences to support these conclusions.

·

Basic reporting

As a Chinese writer, the Chinglish in this manuscript is obvious. The language statement is not refined. There are some repeated sentences can be found both in Result and Discussion. Most of these kinds of language problems I found have been labeled with wavy underline or linear underline in the PDF.

Experimental design

no comment

Validity of the findings

Conclusion are not well stated (See the review tips)

---

## Round 0.2 · Minor Revisions

Please address all the reviewer comments and edits on reviewer #2 pdf, and ensure that the English is corrected by an editing service or otherwise.

Reviewer 1 ·

Basic reporting

Although both reviewers requested that the manuscript should be polished by an English native speaker or English-editing Service Company, there are still many many English grammar mistakes. I strongly suggest the manuscript should be polished before acceptance.

Experimental design

no comment

Validity of the findings

no comment

Additional comments

The revised manuscript has been significantly improved and the authors have addressed my major concerns. However, there are still some problems to be revised.
1. Although both reviewers requested that the manuscript should be polished by an English native speaker or English-editing Service Company, there are still many English grammar mistakes. I strongly suggest the manuscript should be polished before acceptance.
2. Line 17: What is Chinese white Pyrus bretschneideri;
3. Line 20: please use different organs to replace different tissues, because fruit, flower and bud you used in the experiment are organs, not tissues.
4. Line 55: please revise: “the variety of pear cultivars grown is very wide”. I cannot understand the meaning of this sentence. The variety generally refers to Infrageneric taxon. You may mean that there are lots of different pear cultivars grown in China??
5. Lines 55-56: There are several points that need to be clarified or modified. When you use the expression “’Dangshan’ pear (Pyrus bretschneideri)”, you can’t further use Pyrus bretschneideri cv. Dangshan Su. The term “White pear” is often used by Chinese researchers. However, in Asia, there are also other types of pear cultivars or cultivar groups native to different countries: Japanese pear, Chinese sand pear…. Thus you had better use Chinese white pear to replace white pear. Furthermore, if you have read Dr. Teng and his colleague’s papers, you should have known that Pyrus bretschneideri has been misused to refer to the cultivars of the Chinese white pear. They proposed the name for Chinese white pear: Pyrus pyrifolia White Pear Group to replace misused Pyrus bretschneideri. In this paper, you can still use Pyrus bretschneideri. But you are recommended to read papers to understand what is Chinese white pear, what is P. bretschneideri. In addition, what is the meaning of “a characteristic pear type”?
6. Line 58: “The tone cells …. pear's unique taste”? Stone cells are related to flesh texture, not to taste. Please revise the sentence.
7. Line 59: “However, there are few reports on fruit development;” I don’t think so. There are many papers dealing with fruit development in many fruit trees.
8. Line 70: tissues or organs?
9. Line 74: Sweet cherry (Prunus avium) is originated form Europe, Anatolia, Maghreb, and western Asia, not only from Europe.
10. Lines 78-79: the four species included in your paper belong to rose family, but they belong to different subfamily, which means that they don’t have a close evolutionary relationship.
11. Line 367: what is ” the developmental law”?
12. Lines 290-291: the authors attributed different number of IDD genes identified in Malus and Pyrus to origin of the IDD gene family. This may be true. But the authors should also consider the quality of genome sequence of P. bretschneideri used in this study.
13. Acknowledgements: the authors expressed their gratitude to themselves? This is very funny.

·

Basic reporting

The English is largely improved and I have highlighted the words or sentence that are wrong or difficult to understand in the PDF version. The authors have done a lot of effort to improve the quality of the manuscript according to our suggestions, but minor revision is required.

Experimental design

no comment

Validity of the findings

no comment

Additional comments

no comment

---

## Round 0.3 · accepted · Accept

Thank you for addressing the reviewer comments.

# >Because the main focus of the manuscript deals with bioinformatics analysis, and that it focuses on a specific gene family, I would encourage a statement be added, to point to known gene ontology (GO) annotations. Journal manuscripts are often scanned by text-mining software that locates and extracts core data elements, like gene function. Adding standard ontology terms, such as the Gene Ontology (GO, geneontology.org) or others from the OBO fountry (obofoundry.org) can enhance the recognition of your contribution and description. This will also make human curation of literature easier and more accurate.
>
>A couple of questionable items are also listed below:
>
>LINE 74: / de-genomic / . / [Not familiar with this term; can it be re-phrased? Or might it be ‘de novo’?]
>LINE 116: / DnasP / DnaSP /